# The Effect of Psyllium Husk on Intestinal Microbiota in Constipated Patients and Healthy Controls

**DOI:** 10.3390/ijms20020433

**Published:** 2019-01-20

**Authors:** Jonna Jalanka, Giles Major, Kathryn Murray, Gulzar Singh, Adam Nowak, Caroline Kurtz, Inmaculada Silos-Santiago, Jeffrey M. Johnston, Willem M. de Vos, Robin Spiller

**Affiliations:** 1Immunobiology Research Program and Department of Bacteriology and Immunology, Faculty of Medicine, University of Helsinki, 00014 HY Helsinki, Finland; Jonna.Jalanka@helsinki.fi; 2Nottingham Digestive Diseases Centre and NIHR Nottingham Biomedical Research Centre at Nottingham University Hospitals NHS Trust, the University of Nottingham, Notts NG7 2UH, UK; giles.major1@nottingham.ac.uk (G.M.); Kathryn.Murray@nottingham.ac.uk (K.M.); Gulzar.Singh@nottingham.ac.uk (G.S.); 3Nottingham University Hospitals NHS Trust, the University of Nottingham, Nottinghamshire NG5 1PB, UK; adam.nowak@nhs.net; 4Synlogic therapeutics, Cambridge, MA 02142, USA; carolinebkurtz@gmail.com; 5Decibel Therapeutics Inc., Boston, MA 02215, USA; asilos@decibeltx.com; 6Neurogastrx, Inc., Woburn, MA 01801, USA, formerly at Ironwood Pharmaceuticals, Cambridge, MA 02142, USA; jjohnston@neurogastrx.com; 7Laboratory of Microbiology, Wageningen University, 6708WE Wageningen, The Netherlands; willem.devos@wur.nl

**Keywords:** prebiotics, microbiome, ispaghula, constipation, transit

## Abstract

Psyllium is a widely used treatment for constipation. It traps water in the intestine increasing stool water, easing defaecation and altering the colonic environment. We aimed to assess the impact of psyllium on faecal microbiota, whose key role in gut physiology is being increasingly recognised. We performed two randomised, placebo-controlled, double-blinded trials comparing 7 days of psyllium with a placebo (maltodextrin) in 8 healthy volunteers and 16 constipated patients respectively. We measured the patients’ gastrointestnal (GI) transit, faecal water content, short-chain fatty acid (SCFA) and the stool microbiota composition. While psyllium supplement had a small but significant effect on the microbial composition of healthy adults (increasing *Veillonella* and decreasing *Subdoligranulum*), in constipated subjects there were greater effects on the microbial composition (increased *Lachnospira*, *Faecalibacterium*, *Phascolarctobacterium*, *Veillonella* and *Sutterella* and decreased uncultured *Coriobacteria* and *Christensenella*) and alterations in the levels of acetate and propionate. We found several taxa to be associated with altered GI transit, SCFAs and faecal water content in these patients. Significant increases in three genera known to produce butyrate, *Lachnospira*, *Roseburia* and *Faecalibacterium*, correlated with increased faecal water. In summary, psyllium supplementation increased stool water and this was associated with significant changes in microbiota, most marked in constipated patients.

## 1. Introduction

Psyllium husk, derived from the seeds of *Plantago ovata*, consists of highly branched and gel-forming arabinoxylan, a polymer rich in arabinose and xylose which has limited digestibility in humans. However, several members of the intestinal microbiota can utilize these oligosaccharides and their constituent sugars as an energy source [1,2,3,4,5], and therefore, psyllium can be considered to have prebiotic potential. In general, the health-promoting effects of prebiotics include supporting the growth of bacteria beneficial to the host and increasing the production of short-chain fatty acids (SCFA) such as butyrate and propionate previously shown to be positive for colonic health [6]. Another property of psyllium is that it is capable of retaining water in the small intestine, and thereby, increasing water flow into the ascending colon. The resulting increase in the fluidity of colonic content may explain the success of psyllium husk in treating constipation [7,8,9]. In addition to the relief of symptoms through softening of stool and increasing stool frequency, the increase of free water alters the environmental conditions of the colon.

The effects of psyllium husk on host physiology have already been described, however, the effect that it has on the microbial ecosystem has not been well characterised. The effect of psyllium on microbial composition has been studied in an in vitro model mimicking the healthy intestine where it was shown that the microbial composition was not strongly affected. [10] However, there was an indication that the bacterial fermentation of fibres increased colonic levels of SCFAs. Moreover, a recent trial observed that psyllium supplementation protected mice from colitis by reducing the inflammatory response. This response was found to be largely microbiota dependent [11].

In this study, we aimed to determine the effect of psyllium husk supplementation on the intestinal microbiota of both healthy subjects and patients with chronic idiopathic constipation using two separate randomised, placebo-controlled intervention studies. The samples for this study come from two linked, published clinical trials [12] where the effect of psyllium husk on colonic volume and intestinal water content was tested on first healthy adult subjects and then adults with chronic constipation. Here, we built on the results reported previously by concentrating on the microbial composition alterations introduced by psyllium supplementation. We aimed to understand how altered gut environmental conditions induced by psyllium husk affected the microbiota composition. Here, we showed that in healthy controls there was a very small change in the microbiota, however, there was a larger change in the microbial composition of the constipated patients.

## 2. Results

### 2.1. Differences Between Healthy and Constipated Patients at Baseline

The demographics of the participants (study 1: heathy subjects, study 2: adults with chronic constipation) in the two already published clinical trials [12] are shown in Table 1. There were significant differences in age and gender distribution of the participants. Therefore, the effects of psyllium husk on the intestinal microbiota in the two studies were addressed separately and microbial composition differences between the studies were only compared at baseline.

The microbial composition of healthy subjects and constipated patients differed at baseline. The health status accounted for 12% of the total microbial variation (Figure 1, *p* = 0.006). There were 10 taxa contributing to this difference, including 14.0-fold higher abundance of *Desulfovibrio* and 9.4-fold higher levels of *Adlercreutzia* in constipated subjects compared to the healthy controls. Moreover, the levels of *Sutterella* and *Lachnospira* were significantly decreased at baseline in constipated patients (6.7 and 4.6-fold respectively; Table 2). Moreover, there were significant differences in the amount of faecal SCFAs where the constipated subjects had 2.2-fold higher level of faecal acetate than the healthy controls (Table 1).

### 2.2. Psyllium Supplement Has a Small but Significant Effect on the Microbial Composition of Healthy Adults, but no Effect on SCFAs

The temporal stability and resistance to altered environmental conditions are key characteristics of a healthy intestinal microbiota [13]. Dietary supplementation with the carbohydrate inulin has been shown to promote expansion of Bifidobacteria in the stool in two recent studies [14,15]. Inulin has a structure more readily accessible to bacteria for digestion than psyllium, which in comparative studies tends to be less fermentable than gums but more fermentable that cellulose [16]. Therefore, it was not surprising that the composition of relatively few taxa was affected by the psyllium supplementation. The only detected change was a significant increase of *Veillonella* (fold change (fc) = 3.7, *q* = 0.02) and significant decrease of *Subdoligranulum* (fc = −2, *q* = 0.01, Figure 2) with a higher dose of psyllium. There was no significant change from baseline in the levels of SCFAs measured at each study point (Appendix A).

### 2.3. Intestinal Microbiota and SCFAs of Constipated Patients Were Altered With Psyllium Supplementation

In constipated patients, the effect of psyllium supplementation was greater than in healthy controls. There were seven bacterial genera with significantly altered abundance when comparing the time point of psyllium supplementation to other time points (detailed in Table 3). The psyllium husk increased the levels of *Lachnospira* (fc = 1.71), *Faecalibacterium* (fc = 2.71), *Phascolarctobacterium* (fc = 3.62), *Veillonella* (fc = 2.30) and *Sutterella* (fc = 2.13) when compared to baseline, whereas the levels of uncultured *Coriobacteria* (fc = −2.36) and *Christensenella* (fc = −1.83) decreased. Additionally, we detected a statistically significant difference in the amount of acetate and propionate when psyllium was consumed when compared to the baseline levels (Appendix A).

### 2.4. The Impact of Increased Transit on Intestinal Microbiota

One aim of the study was to find associations between altered microbial abundance and whole gut transit time introduced by psyllium consumption. These associations were calculated with linear mixed models, taking into account that there were multiple samples from one subject (detailed in materials and methods). In both study groups of healthy and constipated subjects, an increased amount of *Sutterella* genera was associated with faster transit (Figure 3). Additionally, in the constipated patients, there were altogether 10 additional taxa associated with transit time (Appendix A); these included *Odoribacter* (*q* = 0.03) and *Christensenella* (*q* = 0.05) with negative and positive associations to increased transit time, respectively.

### 2.5. Associations Between Intestinal Microbiota and Faecal Water Content and SCFAs

In line with previous reports, psyllium supplementation increased the participants’ stool water content in both studies [12]. Additionally, we showed that this same measure could be associated with changes in the intestinal microbiota in both study groups. The detailed associations can be found in Appendix A for constipated and healthy subjects respectively. In the constipated patient cohort, we found that several Actinobacteria taxa (genera *Actinomyces*, *Bifidobacterium*, *Asaccharobacter* and uncultured *Coriobacter*) showed significant negative associations with increased stool water content. In addition, three genera known to produce butyrate, *Lachnospira, Roseburia* and *Faecalibacterium*, were significantly increased with increased faecal water content (Figure 4). In the healthy subjects, only *Anaerococcus* showed a negative association to faecal water content.

Short-chain fatty acids are the end products of bacterial fermentation of dietary fibres. We found a significant association between longer transit time and lower levels of SCFAs. The amounts of acetate (cor = −0.59, *p* = 0.0002), butyrate (cor = −0.42, *p* = 0.015) and propionate (cor = −0.59, *p* = 0.0002) all showed a significant correlation with transit time (measured with Weighted Average Position Score (WAPS) after 48 h) in constipated patients. There were no significant correlations between SCFAs and transit times in healthy subjects.

Additionally, we noted that the abundance of 16 bacterial genera were associated with the amount of faecal SCFAs in the constipated patients (Appendix A). A higher amount of acetate was associated with increased levels of two genera from Lachnospiraceaea family, *Roseburia* (*q* = 0.03) and *Lachnospira* (*q* = 0.03) as well as *Faecalibacterium* (*q* = 0.0001; Figure 5). Moreover, the latter two showed positive associations between the levels of propionate (*Lachnospira q* = 0.002, *Faecalibacterium q* = 0.04). In healthy subjects, there were 10 bacterial taxa with associations to the altered levels of faecal SCFAs (Appendix A). These included the positive associations of *Leuconostoc* (*q* = 0.03) with butyrate levels and the positive association of *Blautia* (*q* = 0.07) with acetate levels.

## 3. Discussion

In this study, we utilized two clinical trials to investigate the effects of psyllium husk ingestion on the faecal microbiota composition of both healthy controls and patients with idiopathic constipation [12]. We showed that the environmental changes (change in transit time, faecal water content and the concentration of SCFAs) introduced by the psyllium husk were substantial in both patient groups. However, the healthy subjects showed very little change in their faecal microbial composition, whereas several key members of the microbial ecosystem were altered in the constipated patients.

There were detectable microbial differences between healthy and constipated patients at baseline, including significantly higher levels of *Desulfovibrio* ssp. in the constipated patients. Interestingly, in a recent study, a member of the *Desulfovibrio* genus was shown to reduce transit in an in vivo model [17]. It has also been shown that this taxon produces hydrogen sulphide, which is known to decrease inflammation in the GI tract and to inhibit motility [18,19]. The association of *Desulfovibrio* spp. with constipated patients is a new finding that may be followed up by studies on the relevance of this organism to GI health and its effect on orchestrating motility.

Another genus level taxon that showed significant difference between the study groups was *Lachnospira*, with an over four-fold higher level in the healthy than in the constipated subjects. Additionally, we detected a significant increase of this taxon in these patients after psyllium supplementation and we found positive associations between increased faecal water content and acetate levels. It is known that several members from *Lachnospira* genus are capable of producing lactate and acetate. Lactate may be further metabolized into butyrate or propionate. It has been shown that low concentration of butyrate contributes to constipation by inhibiting mucin secretion [20]. Therefore, decreased levels of these SCFAs produces could contribute to the disease state of constipation.

Psyllium husk has long been used as a treatment for constipation; however, the changes it produces in the composition of the intestinal microbiota have so far not been addressed. We showed that the intestinal microbiota composition of healthy individuals was less affected by the psyllium supplementation than those of the constipated patients. This is in line with previous findings on psyllium supplementation [21] as well as the general understanding where the microbial composition of healthy subjects is regarded to be resistant to environmental changes [22]. It should be noted that the constipated patients had a larger environmental change, as reflected by the change in colonic T1, an Magnetic Resonance Imaging (MRI) time constant reflecting fluidity of the colonic chyme [12], than the healthy subjects; the degree of microbial change may be a reflection of this. We showed that in both studies there was an increase of *Veillonella*, an organism typically more abundant in the small intestine [23]. There were also changes in other small intestinal taxa [24], such as *Sutterella* (belonging to the Proteobacteria phylum), which increased in abundance after psyllium supplementation in the constipated subjects. *Sutterella* bacteria are slow growing organisms [25,26], and therefore, their higher abundance with faster transit could be an indication that these bacteria are washed out of the small intestine and become detectable in greater numbers in faeces. Therefore, we hypothesize that the increase in both of these bacterial taxa is secondary to the supplementation of psyllium and change in the colonic environment rather than increased colonic growth of the bacteria belonging to these taxa.

The supplementation of psyllium significantly increased the numbers of commensals such as *Phascolarctobacterium* [27] and *Faecalibacterium* [28] in the constipated patients. These same organisms have also been shown to have increased abundance after consumption of a polydextrose fibre [29]; *Faecalibacterium* and *Roseburia* have also been previously associated with faster colonic transit [30] and *Faecalibacterium* with loose stools [31]. In our study, the increase in *Faecalibacterium* was also associated with higher stool water content and the increase of both *Phascolarctobacterium* and *Faecalibacterium* with larger amounts of acetate in stools. Interestingly, *Faecalibacterium* converts sugars and acetate into butyrate–an important molecule for intestinal health used by the colonocytes as an energy resource. The favourable change in the acetate concentration may be explained by the increase in abundance of these genera.

Another bacterial genus affected by the psyllium supplementation in the constipated patients was *Christensenella*, which decreased almost two-fold in abundance after psyllium treatment. Previously, *Christensenella* had been associated with hard stools [31], and interestingly, we were able to show a similar trend, since in our study, this genus was negatively associated with increased faecal water content.

In conclusion, we showed that consumption of psyllium husks introduced small but significant changes in the intestinal microbiota of both healthy and constipated patients. This change was more pronounced in the constipated patients. Organisms associated with intestinal short-chain fatty acid production such as *Faecalibacterium* ssp. were increased, an effect that suggests a potential health benefit from psyllium supplementation. Several candidate mechanisms for such an effect were present, such as changes in gut transit and increased intestinal water content. These data support potential health benefits for such mechanisms in addition to their role in the relief of constipation symptoms.

## 4. Materials and Methods

### 4.1. Study Design

This study consisted of two separate, published clinical trials [12] where the effect on psyllium husk on MRI parameters of gastrointestinal physiology was tested on adults without digestive disorders (study 1) and adults with chronic constipation (study 2). The protocols were approved by institutional and national review boards respectively and registered on www.clinicaltrials.gov (NCT01805999 and NCT02144376). All participants gave written informed consent.

A detailed description of these studies has been published previously [12], thus, the methodology is here only given in brief. The study 1, in adults without digestive disorders (*n* = 9; one subject failed to provide stool samples), was a double-blinded, placebo-controlled three-period, three-treatment crossover trial. Treatments were taken in a random order, each taken for a six-day period. Subjects took 14 g of powder three times daily, either 14 g maltodextrin, 14 g Metamucil (providing 7 g psyllium), or a 50:50 mixture 7 g maltodextrin and 7 g Metamucil (providing 3.5 g psyllium). Treatments were separated by washout periods of one week. Four faecal samples were collected, one at baseline and one after each treatment period (Figure 6).

The study 2 included patients (*n* = 16) suffering from chronic constipation, which was defined by meeting the Rome III criteria for either functional constipation or constipation-predominant irritable bowel syndrome. This study was a double-blind, placebo-controlled, two-period, two-treatment study, using only maltodextrin (placebo) and the higher dose (21 g/day) of psyllium. In order to avoid any carryover effects both treatment periods were preceded with ≥10 days of usual laxative use; then, 8 days without laxatives other than rescue therapy. Rescue therapy (oral bisacodyl 5 mg) was permitted in patients who had not opened their bowels for 3 days and were experiencing distressing symptoms, but not in the 48 h before stool sample collection. Four faecal samples were collected: at baseline, after each of the treatments and after the washout period in between treatments (Figure 6).

### 4.2. Measurements of Transit (WAPS)

Subjects swallowed five identical transit markers: cylinder-shaped inert capsules containing 0.4 mL 15 µM gadoteric acid, a positive MRI contrast agent. Ingestion was confirmed in patients by direct observation or via a time-stamped video. The weighted average position score of the transit markers on an MRI scan 24 h after ingestion (WAPS24) was calculated using the formula (sum of the segment number X the number of markers in each segment divided by the total number of segments), as described previously [32]. A higher score denotes slower whole gut transit. WAPS48 was based on the same calculation 48 h after marker ingestion. For healthy controls, transit was measured at the end of each of the three treatment periods and for constipated patients at the end of each of the two treatments.

### 4.3. Analysis from Faecal Samples

#### 4.3.1. Quantification of SCFA by Gas Chromatography–Mass Spectrometry (GC-MS)

0.5 g of faecal sample was suspended in 5 mL 0.5% phosphoric acid (H_3_PO_4_), mixed and centrifuged at 25,000× *g* for 10 min. The SCFA-containing supernatant was filtered through cellulose acetate membrane with pore size 0.2 µm and 700 µL of this supernatant was mixed with 700 µL of ethyl acetate centrifuged 25,000× *g* for 10 min. 300 µL of the supernatant (ethyl acetate layer) was transferred to a clean tube and stored at −80 °C until analysis.

Separation and detection of the short chain fatty acids of interest was achieved using a Trace GC Ultra (Thermo Scientific, Manchester, UK) coupled with a DSQII mass spectrometer (Thermo Scientific). Separations were performed on a Zebron ZB-FFAP column (length 30 m, inner diameter 0.25 mm, and film thickness 0.25 μm; Phenomenex Inc., Macclesfield, UK). The initial oven temperature was set at 60 °C for 1 min then increased at 8 °C min^−1^ to 180 °C. Compound identification was achieved by matching with database mass spectra (NIST/EPA/NIH Mass Spectral Library, Version 2.0d, NIST, Gaithersburg, MD, USA). Identification was further verified by comparing with the retention times and mass spectra of authentic standards. Concentrations of analyte were calculated using ‘Xcalibur’ software (Thermo Scientific, UK). Retention time and specific ions used for quantification are detailed below for each analyte: acetic acid (7.10 min, *m*/*z* 60); propanoic acid (8.41 min, *m*/*z* 57); butyric acid (9.75 min, *m*/*z* 88). The recovery levels were determined on few selected samples; acetic acid 62% (sd = 3.9), propanoic acid 84% (sd = 4.2), butyric acid 95% (sd = 3.5).

#### 4.3.2. Faecal Water

Faecal water was measured by simple drying of faecal samples in a (Jouan, RC10.22, Thermo Scientific, Manchester, UK) vacuum rota-evaporator at 40 °C until constant weight was achieved.

#### 4.3.3. Extraction of the Faecal Microbiota DNA

The participant stool samples were collected and stored at −20 °C at maximum of 24 h and then moved to −80 °C until further analysis. The donated faecal samples (*n* = 92) were extracted according to the validated standard operation procedure, using a combination of mechanical and chemical lysis [33,34].

#### 4.3.4. Analysis of the Intestinal Microbiota

The V4-V5 region of the 16S rRNA gene was analysed with Illumina MiSeq platform. The obtained reads were pre-processed according to the Mare pipeline [35]. In short, the forward reads were trimmed to 150 bps and those with bad quality were discarded. The resultant reads (in average 84,242 reads ranging from 8474 to 189,270 per sample) were clustered to OTUs with UPARSE and chimeras were removed [36]. These were clustered against Silva database [37] at assigning OTUs with 97% similarity. The sequencing data is publicly available in the European Nucleotide Archive (ENA, acc.no. PRJEB29397).

The total microbial load [38] and the abundance of methanogenic archaea [39] determined with qPCR as described previously.

### 4.4. Statistical Analysis

The effect of psyllium was calculated for both trials separately, since there were notable demographic differences between trial populations. Comparisons between subjects from the two trials were only undertaken at baseline to elucidate differences between the groups. The baseline differences in SCFA, stool water content, qPCR results and transit times between the patient groups were calculated either with t-test (parametric data) or Wilcox test (nonparametric data). For microbial diversity, we used the inverse Simpson’s diversity index. Correlations between these data was analysed by using Spearman correlation.

All of the 16S rRNA gene sequencing data were analysed with in-house script from the Mare r-package [35] using R (version 3.2.3). For visualization of the overall microbial composition differences at baseline we performed PCoA using Bray-Curtis dissimilarities. For analysing the effect of health status on the overall microbiota composition, we used a MANOVA test. Differences in bacterial abundances and associations between bacterial taxa and SCFAs, stool water content and transit times within each study were estimated with generalized linear mixed models from all available data points. In both models, the obtained read counts were modelled as a function of time point, using subject ID as a random factor and the total number of reads per sample as an offset, assuming negative binomial distribution. The resultant *p*-values were corrected for multiple comparisons with false discovery rate (FDR), and subsequent *q*-values bellow 0.1 were considered to be significant [40].

## Figures and Tables

**Figure 1 ijms-20-00433-f001:**
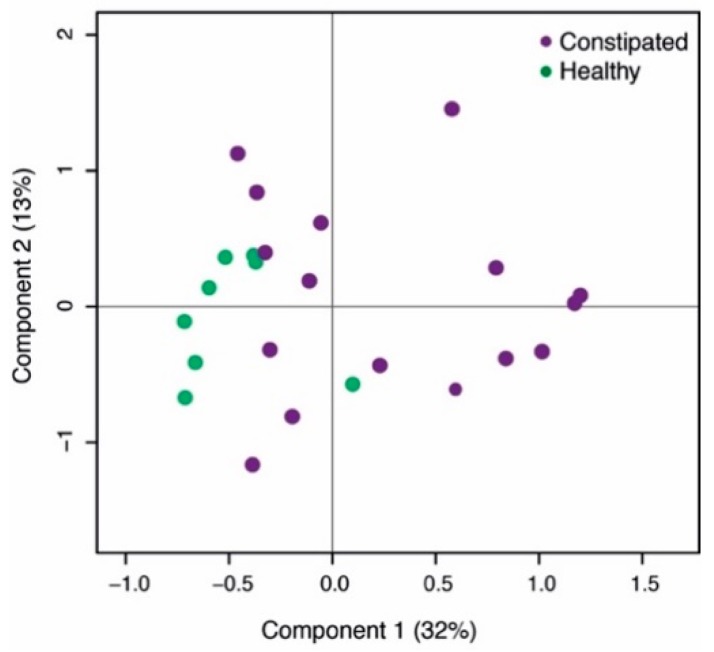
Principal co-ordinate analysis (PCoA) of the baseline microbial differences between the two study populations.

**Figure 2 ijms-20-00433-f002:**
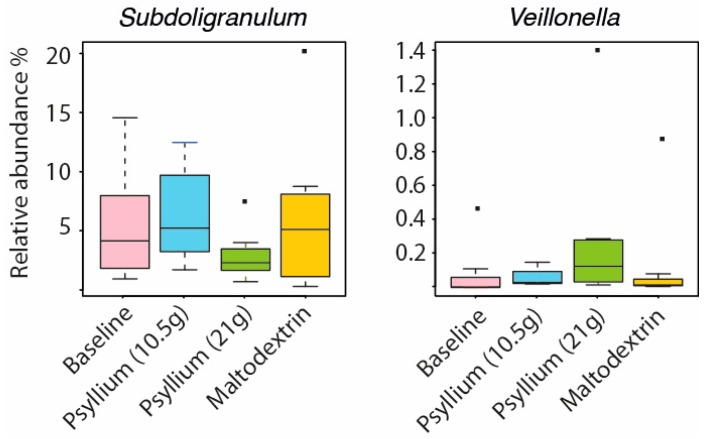
Microbial changes introduced by the psyllium supplementation to the intestinal microbiota of healthy adults.

**Figure 3 ijms-20-00433-f003:**
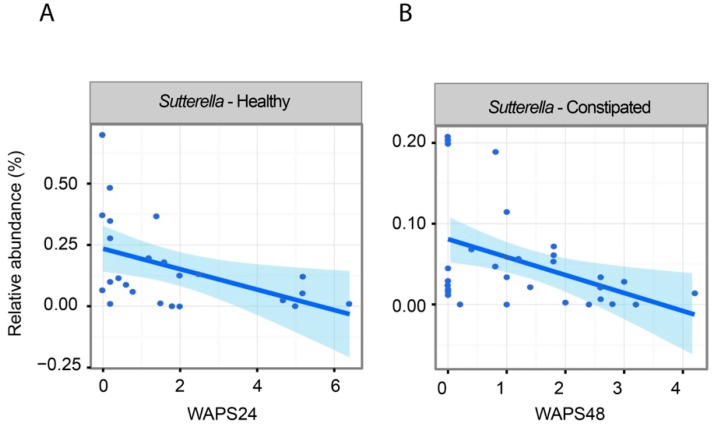
Associations between whole gut transit time and bacterial abundance measured from all available data points: (**A**) relative abundance of genus *Sutterella* and transit time measure (weighted average position score, Weighted Average Position Score (WAPS) measure 24 h after ingestion of transit markers, smaller number indicates longer transit) in healthy subjects; (**B**) relative abundance of genus *Sutterella* and transit (WAPS) measured 48 h after ingestion of transit markers in constipated subjects.

**Figure 4 ijms-20-00433-f004:**
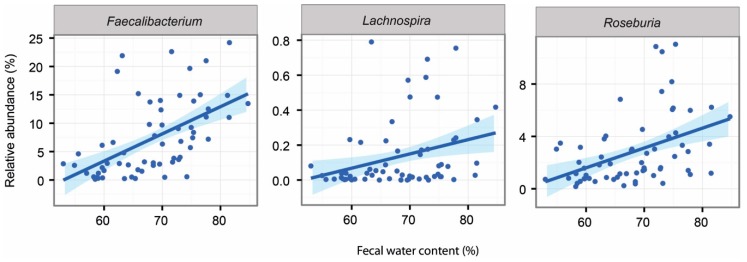
Associations between faecal water content and intestinal genera in constipated subjects at all data points.

**Figure 5 ijms-20-00433-f005:**
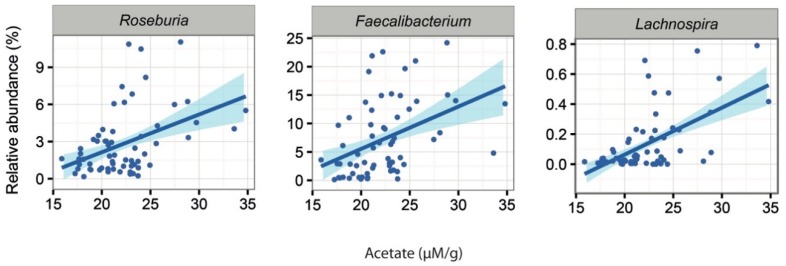
Significant associations between acetate and the abundance of microbial taxa in constipated patients.

**Figure 6 ijms-20-00433-f006:**
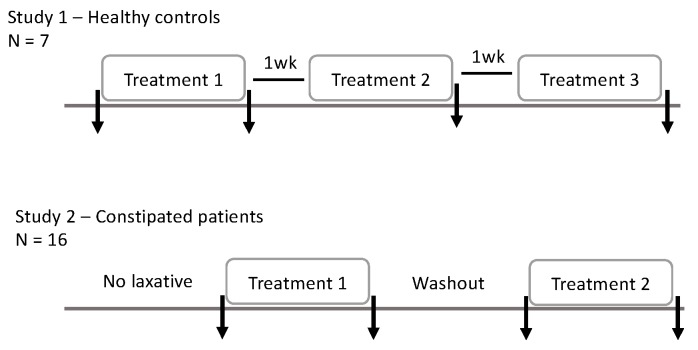
Study outline. The arrows indicate faecal sample collection points.

**Table 1 ijms-20-00433-t001:** The patient demographics and amount of short chain fatty acids and microbial content at baseline. The standard deviations for each number show in brackets.

Demographics	Healthy	Constipated	*p*-Value
Age (y)	25.75 (4.16)	41.00 (15.75)	0.02
Gender (m/f)	6/2	2/14	1.50 × 10^−9^
Weight (kg)	73.05 (12.49)	72.16 (15.65)	0.76
Stool water content (%)	71.13 (5.99)	65.16 (6.55)	0.07
Acetate (µM/g wet faeces)	6.90 (3.72)	21.65 (3.64)	1.5 × 10^−5^
Propanoate (µM/g wet faeces)	5.49 (4.24)	4.70 (3.44)	0.58
Butyrate (µM/g wet faeces)	5.88 (4.39)	5.32 (4.78)	0.78
Total bacterial load (per µL of template log10)	10.47 (0.12)	10.37 (0.28)	0.36
Methanogenic archaea (per µL of template, log10)	4.66 (2.5)	6.21 (2.05)	0.16
Microbial richness	177.75 (74.82)	201.19 (57.35)	0.35
Microbial diversity	8.26(2.71)	10.25 (6.23)	0.40

**Table 2 ijms-20-00433-t002:** Relative abundance of bacterial taxa showing a statistical difference between healthy and constipated patients at baseline. Fc—fold change of taxa in constipated versus healthy subjects.

Taxa	Constipated	Healthy	Fc	*q*-Value
Actinobacteria; Actinobacteria; Actinomycetales; Actinomycetaceae; **Actinomyces**	0.14%	0.02%	5.65	0.01
Actinobacteria; Actinobacteria; Corynebacteriales; Nocardiaceae; **Rhodococcus**	0.03%	0.00%	6.44	0.03
Actinobacteria; Coriobacteriia; Coriobacteriales; Coriobacteriaceae; **Adlercreutzia**	0.10%	0.01%	9.44	0.00
Actinobacteria; Coriobacteriia; Coriobacteriales; Coriobacteriaceae; **uncultured**	0.79%	0.17%	4.52	0.04
Firmicutes; Clostridia; Clostridiales; Christensenellaceae; **Christensenella**	0.13%	0.02%	6.55	0.0002
Firmicutes; Clostridia; Clostridiales; Family XIII; **Incertae Sedis**	0.58%	0.16%	3.52	0.01
Firmicutes; Clostridia; Clostridiales; Lachnospiraceae; **Lachnospira**	0.13%	0.58%	0.22	0.01
Firmicutes; Erysipelotrichia; Erysipelotrichales; Erysipelotrichaceae; **Incertae Sedis**	0.46%	0.06%	7.15	0.01
Proteobacteria; Betaproteobacteria; Burkholderiales; Alcaligenaceae; **Sutterella**	0.03%	0.23%	0.15	0.00
Proteobacteria; Deltaproteobacteria; Desulfovibrionales; Desulfovibrionaceae; **Desulfovibrio**	0.11%	0.01%	14.00	0.01

Bold: effected genus level taxon.

**Table 3 ijms-20-00433-t003:** Bacterial taxa affected by the psyllium treatment of constipated patients. Entries represent average relative abundance. Those time points with statistically significantly different bacterial abundance from the psyllium supplementation are indicated in bold (*q*-values below 0.05).

Taxa	Baseline	Psyllium	Wash-Out	Maltodextrin
Actinobacteria; Coriobacteriia; Coriobacteriales; Coriobacteriaceae; **uncultured**	**0.80%**	0.34%	0.56%	0.88%
Firmicutes; Clostridia; Clostridiales; Christensenellaceae; **Christensenella**	**0.13%**	0.07%	0.08%	0.08%
Firmicutes; Clostridia; Clostridiales; Lachnospiraceae; **Lachnospira**	**0.11%**	0.20%	0.09%	0.13%
Firmicutes; Clostridia; Clostridiales; Ruminococcaceae; **Faecalibacterium**	**3.48%**	9.43%	8.53%	7.47%
Firmicutes; Negativicutes; Selenomonadales; Acidaminococcaceae; **Phascolarctobacterium**	**0.54%**	1.95%	**0.30%**	0.60%
Firmicutes; Negativicutes; Selenomonadales; Veillonellaceae; **Veillonella**	**0.05%**	0.11%	**0.04%**	0.06%
Proteobacteria; Betaproteobacteria; Burkholderiales; Alcaligenaceae; **Sutterella**	**0.03%**	0.07%	**0.02%**	0.03%

Bold: Time points with statistically significantly different bacterial abundance from the psyllium supplementation are indicated in bold.

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
