# Peer review of "The Effect of Psyllium Husk on Intestinal Microbiota in Constipated Patients and Healthy Controls"

_ijms, 2019, doi:10.3390/ijms20020433_

Round 1

Reviewer 1 Report

General Comments

This paper tries to assess the impact of psyllium on fecal microbiota composition using relative data using 16S rDNA sequencing. The research question is relevant & meaningful, but not always clearly stated or to broadly claimed within the manuscript. Results are interesting and novel.

The article is well structured and written in professional English. However, several typing errors have occurred and in some occasions the wrong wording is used, sentences contain redundant information or could be stated clearer. In addition, care should be taken to convey an unambiguous message.

In general, literature references on the topic of psyllium interventions are up to date, yet references on the background could be improved.

Most of the necessary information is provided throughout the text, figures and tables and raw data has been made available in a public database. Figures and tables are of sufficient resolution. One minor issue is that figures sometimes only show a selection of the associations described in the same paragraph without obvious reasons for non-selection.

The authors used the appropriate materials and methods and performed appropriate statistical tests, including multiple testing correction where necessary. The methods are well described and provide enough detail to repeat the analysis.

Major comments

·        62-63: Environmental conditions are indeed altered by increased fecal water. Examples are numerous and include mixing of the luminal content and shear stress (a.o. (https://gut.bmj.com/content/65/1/57). However, although suggested previously (https://gut.bmj.com/content/65/1/57), water activity is a bad example, as recently it has been shown not to be affected by water content in feces (https://gut.bmj.com/content/66/10/1865, Water activity does not shape the microbiota in the human colon).

·        63-65: Altered osmotic pressure is put forward as the main contributor to gut environmental change due to water content fluctuations here, yet this is probably not true and far from proven, as far as I know. I would suggest to stick to ‘altered gut environmental conditions induced by psyllium husk’ or add additional explanation and references to back this up.

·        73-74: I would also suggest to adjust the wording here to ‘altered gut environmental conditions induced by psyllium husk’ or even simply ‘effect of psyllium husk’ on gut microbiota composition. There is no way of disentangling the effects of increased water retention, shorter transit time, or altered SCFA-levels. In addition, the last two could be a consequence of the induced changes in microbiota composition.

·        79: It is currently unclear from the text what has been assessed previously within this study, and what is the additional data acquisition and analysis effort of this study. It might be good to clarify this here and stress what exactly is the novelty of this study (the 16S data, …) in the introduction as well.

·        88: It is stated that health status accounts 12% of the total microbial variation. Yet, as this analysis compares two separate studies, the question arises if health status is wrongly equaled to study-id, which would also include gender, age and sampling effects, or that it was properly dissected from these variables.

·        104-107: These lines touch on an interesting and important point of the study without highlighting it as such: the specificity of prebiotics. Although part of the original definition (refs: papers of Gibson, Roberfroid) there have been suggestions to update this definition regarding the specificity requirements in the light of new evidence (ref: Bindels et al., https://www.ncbi.nlm.nih.gov/pubmed/25824997). Interestingly, the first report on gut microbial changes upon inulin consumption in constipated individuals also showed a specific effect (https://www.ncbi.nlm.nih.gov/pubmed/28213610). It might be worthwhile to further elaborate on this within this paragraph and extend it with the (negative) results on bacteria postulated to grow on psyllium.

·        174: The scope of the study should be narrowed down to the effect of psyllium husk on microbiota composition by including this last part, on microbiota composition, in the sentence.

·        175: The environmental changes referred here should be stated and should be limited to those that were additionally assessed within this study (which is not completely clear as stated before).

·        184: … and speculated to slow down motility. Alternatively, additional references should be provided.

·        192-194: Misses a concluding statement linking this to the results on lachnospira. Message can be made clearer.

·        201: larger environmental change as reflected by colonic transit time, is overgeneralizing. It is more correct, and clearer, to stick to transit time itself. In addition, I suggest the following correction: ‘the microbial changes reflect this’ instead of ‘the microbial change should reflect this’ and addition: ‘given the importance of transit time for the gut microbial ecosystem’ or similar with appropriate references.

·        205: The term specifically is used inappropriately, as the organism was detected in the colon within this study and is thus not specific to the small intestine. I think what the authors were looking for is something like: ‘typically more abundant in’.

·        208: become more prominent in the colonic environment could be interpreted as increased colonic growth, exactly what the authors state they do not support. I would revise this sentence to avoid confusion, e.g. increasing its detection in fecal material. But why would they be washed out from the small intestine?

·        212: stating these commensals as important needs additional explanation andreferences.

·        213-214: Redundant information (This same increase – where the increase of these organisms).

·        226-227: This sentence falls out of the sky and its link with the results could be made clearer.

·        230: in order to state that ‘organisms with potential health effects increased upon psyllium intake’, it should be clarified: 1. which health effects and, 2. which organisms, are meant. This statement is too strong and too general.

·        231: Even if transit acceleration upon psyllium intake is proven, this does not mean it is caused solely by its water retaining properties. There are several other mechanisms described by which prebiotic fibers could increase transit.

·        232: At this moment it is not clear what is under the umbrella of ‘these all’, and this should be clarified.

Minor Comments

In order to improve the manuscript further, I would like to make the following suggestions.

·        Several of the taxa association graphs could be reduced in size without information loss to include the missing associations and give a complete overview. Further, in order to make the reading experience easier, I would suggest to keep the same layout and order in the figure panels if possible, e.g. figure 4 and 5.

·        147: healthy misses an l.

·        148: As Bifidobacterium is one of those bacteria defined as prebiotic since the first reports of this concepts, this result could be compared with available literature on gut microbiota composition with other prebiotic compounds in the discussion.

·        154: subjects in the figure legend misses an s.

·        159: correction: all showed a significant correlation with transit time.

·        160: correction: no significant correlations between SCFAs and transit time

·        182: correction: a member of the Desulfovibrio genus.

·        186: additional evidence of the association with transit time/stool consistency and Desulfovibrio have been published before in population-wide studies.

·        194: correction: … these SCFA could contribute to…

·        197: addition: microbiota composition

·        215: correction: have instead of has.

·        219: misplacement of comma.

·        223: correction: drop decrease.

·        225: correction: show a similar trend

·        226: associated instead of associates.

·         

Author Response

The authors would like to thank the reviewer for his/her work on improving the paper. We have submitted as a separate word document a point by point response to all of the comments. We improved the manuscript like suggested by and hope that the review is satisfied with our responses.

This paper tries to assess the impact of psyllium on fecal microbiota composition using relative data using 16S rDNA sequencing. The research question is relevant & meaningful, but not always clearly stated or to broadly claimed within the manuscript. Results are interesting and novel.

Reviewer 2 Report

The current manuscript is very interesting, demonstrating the effect of psyllium husk consumptions towards health humans and constipated patients. The changes were more significant in constipated patients indicating potential prebiotic efficacies for target functionality i.e. increase gut transit time, water retaining properties and increasing SCFA levels. I think the current manuscript is well written and scientifically sound. Therefore, I would recommend the manuscript to be accepted as it is. 

Author Response

We thank the reviewer for his/her kind comments on the manuscript.